# Improved Spatial Knowledge Acquisition through Sensory Augmentation

**DOI:** 10.3390/brainsci13050720

**Published:** 2023-04-25

**Authors:** Vincent Schmidt, Sabine U. König, Rabia Dilawar, Tracy Sánchez Pacheco, Peter König

**Affiliations:** 1Neurobiopsychology Group, Institute of Cognitive Science, University of Osnabrück, Wachsbleiche 27, 49090 Osnabrück, Germany; vincent.schmidt@uos.de (V.S.);; 2Department of Neurophysiology and Pathophysiology, University Medical Center Hamburg-Eppendorf, 20246 Hamburg, Germany

**Keywords:** spatial navigation, sensory augmentation, cardinal directions, sense of north, pointing, wayfinding, feelSpace belt, virtual reality

## Abstract

Sensory augmentation provides novel opportunities to broaden our knowledge of human perception through external sensors that record and transmit information beyond natural perception. To assess whether such augmented senses affect the acquisition of spatial knowledge during navigation, we trained a group of 27 participants for six weeks with an augmented sense for cardinal directions called the feelSpace belt. Then, we recruited a control group that did not receive the augmented sense and the corresponding training. All 53 participants first explored the Westbrook virtual reality environment for two and a half hours spread over five sessions before assessing their spatial knowledge in four immersive virtual reality tasks measuring cardinal, route, and survey knowledge. We found that the belt group acquired significantly more accurate cardinal and survey knowledge, which was measured in pointing accuracy, distance, and rotation estimates. Interestingly, the augmented sense also positively affected route knowledge, although to a lesser degree. Finally, the belt group reported a significant increase in the use of spatial strategies after training, while the groups’ ratings were comparable at baseline. The results suggest that six weeks of training with the feelSpace belt led to improved survey and route knowledge acquisition. Moreover, the findings of our study could inform the development of assistive technologies for individuals with visual or navigational impairments, which may lead to enhanced navigation skills and quality of life.

## 1. Introduction

According to theories of embodied cognition, there is a reciprocal relationship between action and perception [1,2]. In the context of spatial navigation, body movements lead to new incoming sensory information across the visual, motor, kinesthetic, and vestibular domains [2,3,4]. Changes in the incoming sensory information affect spatial cognition and lead to an updated motor response. The motor response, in turn, changes the sensory perception, and the process repeats until attention is redirected, for instance, toward a different stimulus or stimulus feature (sensorimotor contingencies) [3,4]. Given the reciprocal relationship between action and perception, sensory interaction with the environment appears to be one of the most crucial aspects of spatial knowledge acquisition [5,6,7,8,9].

Spatial knowledge is usually acquired through direct interaction with the environment but can be facilitated through the use of indirect sources such as cartographic maps [10,11,12,13]. To this end, humans combine spatial knowledge acquired from direct experience using an egocentric spatial reference frame [14,15,16] with spatial knowledge acquired through an allocentric reference frame [11,13,17]. During the acquisition of spatial knowledge in a previously unknown environment, people usually first learn the locations of landmarks and connecting routes in terms of left- and right-turn combinations [13,16,18,19,20]. Based on the combination of spatial knowledge acquired from direct experience and additional knowledge from indirect sources, action-relevant representations [3,21] of the environment can be shaped into an organized structure that contains information about the distance and distribution of objects in space [10,11,12]. This process is often referred to as the acquisition of survey knowledge [13,16]. Spatial knowledge acquisition is a multisensory process involving vision, smell, sound, and the vestibular sense [22,23,24,25,26]. More specifically, previous research has shown that the source of spatial exploration influences spatial knowledge acquisition [10]. Thus, spatial knowledge acquisition is a process that depends on various sensory modalities and is affected differently by different sources of spatial information.

Sensory augmentation technologies provide additional sensory information through a native sensory modality and can be used for various usability applications. For instance, several researchers attempted to enhance perception through sensory augmentation of one or multiple senses [27]. Applications of visual sensory augmentation include Augmented Reality (AR) and Virtual Reality (VR) approaches that can be used to present products to customers (i.e., Kabaq) or to support the treatment of various psychological disorders [28,29]. Tactile sensory augmentation revolves around the manipulation or addition of tactile sensory perception. This can be achieved by changing the texture, weight, size, and shape of a perceived object [30,31]. Moreover, technological advances might allow for the development of new ways to stimulate the skin [27]. Auditory sensory augmentation, for instance, playing sounds through headphones, can be used to enhance and potentially even alter the quality of a perception perceived through a different modality [32,33,34]. Multisensory augmentation technologies involve the augmentation of at least two sensory modalities. For instance, the perception of a liquid can be augmented through a device that combines tactile vibration with sounds [31,35]. These findings suggest that augmented senses can be learned and integrated together with native senses.

While existing sensory experiences can be enhanced, sensory augmentation also allows creating entirely new sensory experiences. In the context of spatial navigation, tactile sensory augmentation devices that provide access to previously unknown information have been suggested as a way to boost spatial navigation. For instance, the “EyeCane” is a cane that offers blind users a novel sense of distances toward objects in space [36,37,38]. It uses an infrared signal to measure the distance toward a pointed object and produces a corresponding auditory signal. Moreover, tactile sensory augmentation devices have been developed to study how users learn to use a new sense [39,40,41,42]. A sensory augmentation device that might be particularly interesting with regard to the study of spatial knowledge acquisition is the feelSpace belt. The feelSpace belt gives the user a sense of cardinal directions through tactile stimulation around the waist, pointing to the direction of cardinal north (for a detailed description, see Section 4.2.2). Previous research on the feelSpace belt found that users reported an altered perception of space and a subjectively improved spatial navigation ability after wearing and training with the belt for an extended duration of six weeks [41]. Moreover, sleep-electroencephalography (EEG) has shown procedural learning to be increased during the first week of training with the device [43]. Further investigations using functional magnetic resonance imaging (fMRI) have shown differential activations in sensory and higher motor brain regions during a virtual path integration task [43]. Based on these subjective and physiological reports, the feelSpace belt appears to be a useful tool to study how humans attain a novel sense of cardinal directions.

Following this line of research, we are interested in the behavioral effects induced by the feelspace belt. König and colleagues [43] reported that no behavioral effect for the feelSpace belt could be found in a complex homing paradigm. A potential explanation is that the complex homing paradigm was conducted on a very small scale (19–22 m) [43] and could therefore have been solved by exclusively using the vestibular system [44,45]. Moreover, these studies were challenged with small sample sizes, for instance, due to time-intensive experimental procedures [41,43] or a focus on specific test subjects [40,46]. Interestingly, previous research investigating spatial knowledge in real-world environments identified the challenge that there is a large interindividual variability in spatial navigation ability and the acquisition of spatial knowledge across the population [47,48,49], which increases even more when factors such as the degree of familiarity with the environment are likely to impact the results. These are merely impossible to account for when conducting experiments in large-scale real-world environments [50,51,52]. Overall, previous research has emphasized the importance of a high degree of experimental control, which can be achieved by providing a sufficient but not excessively large environment, a reasonable sample size to be able to detect group differences statistically, and finally, by ensuring that participants’ familiarity with the environment is comparable at baseline.

By allowing for the creation of a large amount of new, fully customizable environments with the possibility to track variables such as body or eye movements precisely, VR appears to be a promising solution to increase experimental control while maintaining a high degree of realism. As connoted earlier, real-world spatial navigation depends on a wide variety of variables that change constantly and are difficult to control or predict. Sacrificing experimental control for ecological validity could increase the risk of including systematic bias, such as pre-existing differences. Consequently, experimental paradigms within the field of spatial navigation often include highly controlled environments with limited exploration and movement possibilities, such as maze navigation tasks [53,54,55,56,57,58,59,60]. In contrast, recent studies suggested using visually realistic but nevertheless controllable VR environments [10,61,62,63,64,65]. Previous studies that used VR to study spatial navigation reported promising results [66]. Nevertheless, a previous VR study on the feelSpace belt showed that the approach requires sufficient exploration time in the unknown environment as it could otherwise not be distinguished from chance-level performance [67]. Given that adequate exploration time is provided, urban VR environments might offer a realistic yet controllable alternative to study spatial navigation.

Here, to account for the aforementioned challenges in investigating the effect of the feelSpace belt on spatial learning, we investigate the effect of real-world training with the feelSpace belt on spatial navigation and spatial knowledge acquisition in a custom-made naturalistic urban VR environment. To this end, we recruited a sufficiently large sample of participants and trained them to use the feelSpace belt as an augmented sense for cardinal directions. First, participants wore and trained with the belt for six weeks during their everyday life. Then, participants individually explored a large-scale urban VR environment in five separate 30-min sessions while wearing the feelSpace belt indicating the predefined virtual direction of north. Eventually, we assessed four tasks, each measuring different aspects of knowledge of cardinal directions, survey knowledge, or route knowledge. In order to compare the performance of the belt group, an equally sized control group also completed the exploration and test sessions in the VR environment (see Figure 1). By comparing the spatial knowledge of the belt and the control group after completion of the task assessments, we aim to answer whether tactile sensory augmentation with the feelSpace belt, by providing a sense of cardinal north, supports spatial knowledge acquisition in a realistic urban VR environment.

## 2. Results

In order to answer the research question of whether sensory augmentation with the feelSpace belt, by providing a sense of north, affects spatial navigation strategies and performance, 56 participants were recruited and divided across two groups. The belt group consisted of 29 participants that were provided with an augmented sense of cardinal north in the form of the feelSpace belt. The control group consisting of 27 participants was not provided with the augmented sense and was therefore limited to using their native senses. Due to unforeseen time constraints and other unknown reasons, three participants did not complete the study and were excluded. This resulted in a final sample of 53 participants (27 belt group). The data from all 53 participants that completed the experiment were considered for the analysis.

### 2.1. Baseline Comparison of Spatial Strategy Use

A Wilcoxon rank sum test for independent samples was used to compare the groups’ scores on the FRS subscales [68] (egocentric/global, survey, or cardinal directions navigation strategies) at baseline (see Section 4.3). The results showed no significant difference between the two groups at baseline across all three scales, *W*s(*n*_belt_
≤ 270, *n*_control_
≤ 260) ≤ 36,004, *p*s > 0.05 (see Figure 2). Thus, there is no indication that the belt and control groups were different regarding their reported use of spatial navigation strategies before the experimental procedures.

### 2.2. Training Phase of the Belt Group

The experimental group participants were asked to fill in a daily and a weekly questionnaire (see Appendix A) that assessed, for instance, their belt-wearing and active training times. Results of the training times showed that participants, on average, wore the belt for 9.18 h (h) per day (d; *s* = 2.78). The self-reported active training time was on average 2.27 h/d (*s* = 1.74). Compared to a previous study by Kaspar and colleagues [41,43] that found active training time (*M* = 1.57 h/d) to be correlated to perceiving the belt positively, the belt group in our current study trained for about 42 min more per day. At the same time, the average belt-wearing time was slightly lower than in the previous study (*M* = 10.49 h/d). Overall, the training time slightly exceeded the target values and the values reported in previous studies, showing participants’ compliance with the experimental procedures.

We assessed the FRS questionnaire a second time after the belt group completed the training phase to investigate whether the training with the augmented sense affected the use of spatial navigation strategies. Compared to the baseline, the median score of the participants in the belt group scored was unchanged on the ‘egocentric/global orientation’ scale (see Figure 2; *W*s(*n*_belt_
≤ 270, *n*_control_
≤ 260) ≤ 36,004, *p*s > 0.05). It increased by one on the ‘survey’ scale and by two on the ‘cardinal directions’ scale. For a statistical evaluation, we focus not on the median but test the full distributions using a Wilcoxon signed rank test for dependent samples. The rating on all three scales differed significantly after training as compared to the baseline measurement (egocentric scale, *W*(*n*s = 270) = 3492, *p* < 0.001; survey strategies, *W*(*n*s = 189) = 1232, *p* < 0.001; and cardinal directions, *W*(*n*s = 54) = 21, *p* < 0.001). Analysis of Wilcoxon effect sizes (*r*) revealed large effects for the ‘survey’ (*r*(*n*s = 189) = 0.503) and the ‘cardinal directions’ (*r*(*n*s = 189) = 0.733) scale, and a moderate effect size for the ‘egocentric’ scale (*r*(*n*s = 270) = 0.404). Taken altogether, the training with the belt caused significant changes on all scales, with larger effect sizes on the ‘survey’ and ‘cardinal directions’ scales than on the ‘egocentric’ scale.

### 2.3. Task Assessment Phase

Following the VR exploration sessions (Section 4.4), we assessed participants’ spatial navigation performance using four spatial tasks: pointing north, pointing to building locations, building placement, and wayfinding. The first task directly addresses knowledge of cardinal directions. The second and third tasks measure different aspects of survey knowledge (see Section 4.5). The fourth task, wayfinding, relies on left- and right-turn combinations, commonly referred to as route knowledge, related to an egocentric perspective. To assess the effect of the augmented sense, the belt group wore the belt during all tasks but pointing north.

#### 2.3.1. Pointing North

The pointing north task assessed participants’ knowledge of cardinal directions. They were placed in front of one of the buildings with street art on their walls (for a detailed description, see Section 4.5.1) and were asked to point in the direction of cardinal north from this location. This procedure was repeated fifty-six times in a randomized order which resulted in fifty-six unique trials, each repeated once. Note that the belt group performed this test with a deactivated belt. Moreover, to test the absence of indirect cues for cardinal directions (Section 4.5.1), the control group was left unaware of the north direction. During each trial, the pointing accuracy was measured in terms of the pointing error from the true beeline direction of cardinal north in the VR environment (see Figure 3). The pointing error was calculated as the absolute angular difference (0–180°) between the participant’s pointing direction and the beeline vector pointing from the participant’s location toward the north. The belt group performed significantly better than the control group with a difference of the median error of 79.94° and a large Wilcoxon effect size (*r*(*n*_belt_ = 3019, *n*_control_ = 2912) = 0.526). The distribution of pointing errors was significantly different between the two groups, as revealed by a Wilcoxon rank sum test (*W*(*n*_belt_ = 3019, *n*_control_ = 2912) = 1,727,085, *p* < 0.001). The results demonstrate that participants in the belt group acquired knowledge about cardinal directions, which was available even while the belt was deactivated.

#### 2.3.2. Pointing to Building Locations

The pointing to building locations task assesses aspects of the participants’ survey knowledge. They had to point in a beeline from their current location toward a given building located in the city. Therefore, participants were placed in front of eight building locations. From each, they pointed toward the remaining seven before being teleported to the next building location (for a detailed description, see Section 4.5.2). In total, the task consisted of fifty-six trials, each repeated once. Similar to the pointing north task, the pointing error was calculated as the absolute angular difference between the participant’s pointing direction and the beeline vector pointing from the participant’s location toward the actual building location. The belt group performed better than the control group in pointing toward given building locations with a difference of the median error of 9.83° (see Figure 4) and correspondingly small Wilcoxon effect size (*r*(*n*_belt_ = 3019, *n*_control_ = 2912) = 0.101). A Wilcoxon rank sum test considering the full distributions revealed that the difference between groups was statistically significant (*W*(*n*_belt_ = 3013, *n*_control_ = 2912) = 3,873,130, *p* < 0.001). These results show that the belt groups’ accuracy in pointing toward building locations was higher, indicating an increased acquisition of survey knowledge.

#### 2.3.3. Building Placement

The building placement task extends the assessment of survey knowledge to include metric aspects. Participants were instructed to place 28 buildings back in their original location in the city. Then, they were placed at one location in the city from which they had to position a building mapped to their controller in the correct direction, distance, and orientation (for a detailed description, see Section 4.5.3) from a first-person perspective. Thereby, we measured three different outcomes: placement direction error, distance error, and rotation error. First, we analyzed the placement direction error (°) in the same way we analyzed the pointing error in the pointing to building locations task. Then, we additionally analyzed the distance to the correct location and the rotation error for the building placement. To this end, we calculated the distance between the placement location and the correct target location in the virtual equivalent of meters (m) and the angle of deviation (°) from the correct building rotation (facing direction). As the three outcome variables might rely on different aspects of spatial navigation, they were considered separately for the analysis. Compared to the control group, the belt group placed the houses with a median error smaller by 8.09° (see Figure 5A). Similarly, the participants in the belt group were more accurate than the controls in reproducing the initial building orientation, with the median error reduced by 27.63° (see Figure 5B). Finally, the belt group was significantly more accurate than the control group in placing the building at the correct distance, with the median error reduced by 21.70 m (see Figure 5C). Wilcoxon effect sizes calculated for all three outcome variables indicate that the effect sizes were small (*r*s(*n*s_belt_ = 744, *n*s_control_ = 724) ≤ 0.156). For statistical testing, three separate Wilcoxon rank sum tests were performed, each analyzing the performance differences between the belt and the control group concerning one of the outcome variables. The results reveal a significantly higher accuracy in the belt group regarding all three outcome variables as compared to the control group (*W*(*n*_belt_ = 744, *n*_control_ = 724) = 246,938, *p* < 0.001, *W*(*n*_belt_ = 744, *n*_control_ = 724) = 220,824, *p* < 0.001, *W*(*n*_belt_ = 744, *n*_control_ = 724) = 224,776, *p* < 0.001; placement direction, building orientation, distance respectively). In conclusion, the belt group placed the buildings more accurately than the control group regarding the buildings’ direction, distance, and orientation, indicating an improvement in survey knowledge.

#### 2.3.4. Wayfinding

Finally, we assessed route knowledge through the wayfinding task. Starting at a designated location in the city, participants had to find the most efficient path to a given building location (for a detailed description, see Section 4.5.4). The task was conducted to determine whether the augmented sense of cardinal north would lead to improved wayfinding. In order to complete the task, each participant navigated toward nine different locations in regular or reversed order. Each trial had a maximum duration, and trials where the target location was not reached within this time were set to the maximum duration of the longest path. The task aims to provide a close-to-real-life application of spatial navigation strategies. Compared to the control group, the participants of the belt group completed this task faster by 43.89 s (difference of medians, see Figure 6), which reflects a small Wilcoxon effect size (*r*(*n*_belt_ = 243, *n*_control_ = 234) = 0.120). A Wilcoxon rank sum test revealed that this difference between the groups was significant (*W*(*n*_belt_ = 243, *n*_control_ = 234) = 24,552, *p* < 0.01). The results show that participants in the belt group were more efficient in finding routes than control participants.

## 3. Discussion

The research question we aimed to answer was whether tactile sensory augmentation with the feelSpace belt supports and improves spatial knowledge acquisition by providing a sense of cardinal north. After training with the belt, participants subjectively rated their navigation strategies higher on all scales of the FRS questionnaire. The effect size was larger concerning the survey and cardinal direction scale than the egocentric scale. The behavioral results of the spatial VR tasks demonstrated improved pointing accuracy and performance in all aspects of the building placement task. Furthermore, a small improvement was demonstrated in the route-finding task. This pattern of results is fully compatible with a noticeable improvement in survey knowledge and a modest improvement in route knowledge. As such, the match of subjective and behavioral results presented here is a critical keystone demonstrating the potential of tactile sensory augmentation when it is practically applied to spatial navigation. The present study rests heavily on virtual reality techniques to achieve several objectives. Firstly, VR techniques provide full control over the design and the corresponding size of the environment. Compared to previous studies investigating the feelSpace belt, as well as some studies of general spatial cognition, the virtual city was larger and thereby allowed more extensive exploration [51,52,62]. The vestibular system informs about rotations on a time scale of not more than 30 s [69], which is of limited use in such large environments. In contrast, on a large spatial scale, the feelSpace belt provides reliable information throughout the exploration session [43] (also see Section 4.2.1). Secondly, previous studies showed that it is essential to account for the participant’s familiarity with the environment [50,70]. Using a custom-designed VR city gives precise control of participants’ familiarity and spatial knowledge of the environment [71,72,73,74]. Third, a six-week training phase provided sufficient time for the participants to accustom and master the processing of the information provided by the belt [39,41]. This is crucial since previous studies provided evidence that using the feelSpace belt without training rendered no behavioral effects in spatial virtual reality tasks [67]. Fourth, by increasing the exploration sessions to five [10,62], we further ensured that participants were reasonably familiar with the VR environment, reaching a performance clearly above chance levels [67]. Fifth, previous studies used bodily rotations [10] and, in some cases, physical movement—for instance, using omnidirectional treadmills [75,76]—as movement input. In the current study, physical full-body rotation movement was translated to the VR environment, which Riecke and colleagues [6] argued might be sufficient to gain considerable navigation improvements. The physical body rotation was then translated by a controller to induce movement. More specifically, in contrast to previous studies, the translation of the movement input was relative to the direction of a chest-mounted body tracker. The direction of the body tracker was translated as the forward direction for movement input. Most contemporary VR systems translate movement inputs from a controller relative to translation in the direction of the head/gaze. As participants might not always look into the direction they want to walk toward, or the other way around, this can be disturbing. The body tracker mounted around their chest allows for intuitive free viewing without constant adaptation of movement inputs. Finally, we increased the sample size based on the outcome of a power analysis, thereby increasing the likelihood of a given effect showing up statistically. Overall, in the present study, we successfully addressed a multitude of problems associated with studying spatial navigation in VR.

This does not mean that VR techniques are without problems. While VR allows for the creation and presentation of realistic and immersive visual environments [61,63,64,65,66], the environments are less information-rich and often less interactive than real environments. First, the city is relatively sparse for the participants as there is no traffic or life. To address this issue, we currently investigate the spatial exploration of virtual cities that encompass human avatars. Second, it contains several buildings and stores, but working, shopping, or meeting other people in Westbrook is not possible. The relatively sparse environment and a lack of interaction possibilities could have negatively affected participants’ motivation to explore the environment further or could have led to a different exploration style compared to the real world. We tried to compensate for this by placing graffiti on all buildings that would later be used in the task and by giving participants a goal to explore: take photographs of the graffiti. Third, physical movement speed can vary from slow walking to running depending on the situation and individual preference of the agent. Interestingly, previous research has shown a tradeoff between movement speed and accuracy [77,78,79]. Movement speed in the virtual environment was limited to a maximum of 5.1 km/h, which is the estimated preferred average movement speed of humans [80,81,82]. While this resulted in more methodological consistency and presumably higher movement accuracy across participants, it might have negatively affected participants who prefer higher movement speeds, such as people living in large cities [83,84]. Fourth, hardware limitations such as tracking inaccuracy could have led to involuntary or delayed movements, which cause a mismatch between action and perception. Involuntary movements are associated with an increased occurrence of cybersickness [85]. To limit the risk of the occurrence of tracking issues, our experimental setup (also see Section 4.2.3) included four lighthouses, each tracking the controller input and the participant’s movement from one side. Fifth, real-world environments are dynamic and multisensory [50,86]. While vision and movement can be transferred into the VR environment reasonably well, as described above, other senses, such as smell, also appear to play an essential role in spatial navigation. For instance, several studies using virtual reality found a simulated smell to improve the memory of places or objects in the environment [87,88] while also increasing the feeling of presence in the environment [89]. At the same time, further technological advances might enable an even more realistic and immersive visual presentation of VR environments. Finally, previous research has emphasized that spatial knowledge can be transferred into a different spatial frame of reference [12,90]. Therefore, it cannot be excluded that an increase in one type of spatial knowledge, for instance, route knowledge, could derive from an increase in another higher-level type of spatial knowledge, such as survey knowledge. Although the addressed issues are unlikely to affect or change the results reported in this study systematically, they are crucial factors to consider to further improve the design of the VR tasks for future spatial navigation research. Although random participant assignment was not feasible in our study due to significant differences in subject involvement and varying timelines, we administered the spatial strategies questionnaire to the experimental and control groups at the start of the study. Our analysis revealed no significant differences in all three questionnaire scales between the two groups. However, after training, we observed consistent distinctions between the pre and post-training scores of the belt group. While a confounding bias cannot be entirely ruled out, our testing suggests no indication of such a bias. In line with previous research [47,48,49,50,51,52], our results show that there was a large interindividual variability between participants in the same group (see Figure 3, Figure 4, Figure 5 and Figure 6). Some participants, especially in the belt group, appear to solve the tasks consistently well. In contrast, others are unable to solve the task, as reflected by a median performance around the chance level. Consequently, the Wilcoxon effect sizes [91] in the pointing to building locations, building placement, and wayfinding tasks are small, although the median difference appears to be substantial when considering absolute units. Furthermore, the sample was drawn mostly from university-internal recruitment through mailing lists. Therefore, the study likely attracted people naturally interested in spatial cognition or related research topics. While all participants were unaware of the hypothesis being tested, the study did not follow a double-blind procedure. Studies not following double-blind procedures are particularly susceptible to bias [92]. Specifically, the fact that the belt group engaged in six weeks of training with the belt leading up to the experiment while the control group did not might have biased the results. In contradiction to this assumption, the reported active training time with the belt was higher than in previous studies. More interestingly, we found no significant difference between the belt and control groups in terms of spatial strategy use before the study. While this finding does not exclude that there were pre-existing differences or biases compared to the general population, it supports the validity of the comparison of spatial knowledge between groups after the VR exploration phase.

### 3.1. The Effect of the feelSpace Belt on Spatial Knowledge Acquisition

In order to evaluate the effect of the feelSpace belt on spatial knowledge acquisition, we assessed cardinal, survey, and route knowledge in four VR tasks. Firstly, we discuss the effect of the feelSpace belt on cardinal directions through the pointing north task. Secondly, we discuss the effect of the feelSpace belt on survey knowledge assessed by pointing to building locations and building placement. Finally, we discuss a potential effect on route knowledge as reflected by the results of the wayfinding task. Taken altogether, the results suggest that the belt group acquired significantly more cardinal, survey, and to a lesser degree also route knowledge.

The pointing north task assessed participants’ knowledge of the cardinal directions. It was based on König and colleagues’ [10,50,62] “absolute orientation task” that assessed participants’ knowledge about the direction of cardinal north using a two-alternative forced choice task design. While the “absolute orientation task” was conducted on a computer screen and participants were not immersed in the VR, the present pointing north task allows for dynamic measurements of pointing direction directly inside the VR environment. The results of the pointing north task align with the reported increased use of spatial strategies related to cardinal directions [68], indicating that more cardinal spatial knowledge has been acquired. A previous study has shown a similar effect after multiple weeks of training with the feelSpace belt [41]. Indicating the cardinal north is beneficial, as it provides a global orientation point that can be used to gain knowledge about the alignment of small spaces [50,62]. Upon successful alignment, these smaller snippets can then form an integrated large-scale survey knowledge [93]. Overall, the pointing north task results show that sensory interaction through an augmented sense of cardinal directions facilitates spatial knowledge acquisition related to cardinal directions.

The pointing to building locations task aims to assess survey knowledge. Tasks similar to the pointing to building locations task have previously been used for the same purpose [94,95]. Moreover, Berry and Bell [96] found that survey knowledge was more predictive of pointing accuracy than route knowledge. The pointing to building locations task results indicate that the augmented sense led the belt group to acquire more survey knowledge [13,16,19,20]. These results support the hypothesis that the training with the augmented sense in the natural environment followed by the exploration of the VR with the belt improved the acquisition of survey knowledge. However, several studies argued that the ability to point in the general direction of a building could not be equated with the ability to use a cognitive map. Indeed, the pointing direction to a building and the initial path segment of a route to that building are not statistically independent. More often than not, the optimal route will be offset at an angle of less than 90° relative to the pointing direction than at larger angles. For this reason, other tests of survey knowledge have been developed. For example, survey knowledge is evaluated when comparing a map drawn by a participant to the real map of a given environment [97,98,99,100,101]. However, several researchers have argued that improved performance in a map-drawing task could also derive from an upscaled route representation [12,102,103]. Furthermore, drawing a map requires a perspective switch from an egocentric perspective to a bird’s eye view. Other tasks, such as finding shortcuts [104,105,106,107] or estimating distances [95,100,101,108], have also been used to evaluate survey knowledge. These tasks, however, are also not without problems, as they allow the gathering of additional information during the performance of a task (shortcut) or are particularly susceptible to distortion [109,110] and large pre-existing individual differences [95,111]. The building placement task aimed to combine the advantages of performing an efficient test within VR and addressing survey knowledge more directly than estimating a direction. Specifically, it tests a metric property in the form of the distance and the relative orientation of the building. These are two aspects that are also relevant in map-drawing tasks. However, the building placement task does not require a perspective switch. The results from the building placement task align with the results of the pointing to building locations task regarding placement or pointing direction accuracy, respectively. Moreover, the results align with previous studies showing increased reported use of navigation strategies related to survey knowledge over time [41,43]. The finding that belt participants also performed more accurately at estimating distances and orientations of buildings in the VR environment further supports the hypothesis that the feelSpace belt improves the acquisition of survey knowledge after training and VR exploration [13,16,19,20].

Finally, the wayfinding task was used to assess route knowledge. The wayfinding task is common in spatial navigation research, especially in maze studies [53,54,55,56,57,58,59,60]. As compared to artificial environments such as mazes which usually incorporate a complex structure of paths, the wayfinding task in this study was applied to an environment that is complex due its size and variety. The results indicate the feelSpace belt and the corresponding training also lead to improved wayfinding, which reflects increased route knowledge acquisition [112,113]. In summary, the results reveal that belt participants were able to acquire more route knowledge, allowing them to find their way more efficiently than control participants.

### 3.2. Sensory Augmentation Can Enhance Human Performance

The effect of the feelSpace belt after training and VR exploration with the belt on spatial knowledge acquisition demonstrates that sensory augmentation devices are an effective way to enrich human biological sensory perception with additional information when participants had the chance to incorporate the augmented sense through previous training. In the present study, we show that the combination of a six-week training period, exploring the VR environment with the belt, and performing the VR tasks with the belt (except pointing north) is sufficient to achieve the reported behavioral benefits. This does not mean that all components of that combination are necessary, as they could potentially be replaced. However, previous studies [41,43] had a comparable six-week training period for both groups except for wearing the belt and showed differences in the quality of perception and physiological results obtained by EEG and fMRI, but they did not show a significant behavioral improvement. This indicates that a six-week training period by itself is not sufficient. Additionally, follow-up studies exploring a VR city without the belt [10] and with the belt but without previous training in a natural environment [67] did not indicate significant differences in pointing accuracy. This shows that using the belt by itself is not sufficient. Therefore, we assume that the improved spatial knowledge reported in this paper is likely a result of the combination of training with the augmented sense in the real world, exploring the VR environment with the belt, and performing the VR tasks with the belt which together render a sufficient combination for behavioral improvement. We propose that the constant availability of congruent information on cardinal north during training provides an accessible and reliable spatial reference frame, contributing to the positive effect on spatial knowledge acquisition. However, whether the tested combination of training with the belt and using the belt in VR is the minimal sufficient, i.e., necessary set, remains to be shown.

The current findings support the theory of sensorimotor contingencies (SMCs) [3,4] showing that law-like relations between a previously unknown signal within the environment and a known set of behavioral responses can be learned and applied effectively. While some SMCs have been shown to already be present during childhood [114,115,116], the current study contributes to the growing scientific evidence showing that healthy adults can also effectively learn to interpret a novel sensory signal [39,117] after a training period. In this context, this study provides evidence that the learned SMCs can lead to significant performance improvements in humans when they are applied to a meaningful context. Similar techniques could be applied to sensory substitution, for instance, to compensate for sensory disability, disorder, or deficits [40,118,119,120,121,122,123,124], or to other sensory augmentation devices to further enhance human perception by allowing humans to perceive information that they cannot naturally incorporate through their senses [27,30,31,32,33,34,35]. Specifically, similar sensory augmentation techniques have been used to aid deaf and blind people during spatial navigation, showing that training with sensory augmentation devices can lead to improved navigational performance and reduced discomfort [46,121,125,126].

Nevertheless, there are various future challenges regarding the translation of information recorded by one or multiple sensors to a meaningful sensory signal that humans can perceive. For many types of information that can be recorded through sensors, it remains unclear how the recorded signal could be translated to a native sensory modality in an intuitive but meaningful way. Consequently, there is a need to investigate further which sensory modalities and, more specifically, which combinations of sensory modalities and signals are suitable to support the formation of SMCs. Furthermore, it remains to be investigated whether and how more complex signals, for instance dynamic information, can be augmented or substituted to support the formation of SMCs in humans. While the use of a linear signal was essential to the aim of the current study, the expression of dynamic information would require the creation of meaningful dynamic output signals, for instance, using customizable frequencies and levels of intensity [127]. The current study demonstrates that healthy human adults can create new SMCs between their bodily movements and a previously unknown augmented sense, yet the limitations of sensory augmentation are currently unclear.

### 3.3. Conclusions

We conclude that using the feelSpace belt after training in a natural environment and exploration of the VR environment supports spatial navigation in a realistic VR environment by providing a sense of north. First, we conclude that changes to the VR environment and the VR movement system based on previous studies improved the quality of the acquired data. Previous results on self-reported changes in space perception [41,43] could be reproduced, expanded, and supported with behavioral results. In this line, the feelSpace belt has been shown to facilitate the acquisition of knowledge of cardinal directions, survey knowledge, and to a lesser degree, also route knowledge perceptually and behaviorally. Nevertheless, interindividual variability is large [47,48,49]. Therefore, future research could investigate interindividual differences in spatial knowledge acquisition. The VR spatial navigation tasks we created resolve a variety of issues reported by previous studies to assess the acquisition of survey and route knowledge more accurately. Based on the promising effect of the augmented sense on space perception, it is evident that sensorimotor contingencies can be formed between an augmented sense and a previously unknown environment. Future research on sensory augmentation could focus on employing electrophysiological measures such as EEG to assess what cognitive processes are involved in processing the augmented signal of cardinal north.

## 4. Materials and Methods

The study followed a between-subject mixed-methods design. The general aim of the study was to compare behavioral and qualitative self-reported changes related to spatial navigation between two groups: the belt group that wore and trained with the feelSpace belt for six weeks, during the VR exploration sessions, and all but one task (not in the pointing north task), and the control group that performed the VR exploration sessions and the experimental part, not wearing the feelSpace belt at any point in time. By choosing a between-subject design, we can analyze the differences between the belt and a control group. Moreover, we increase methodological consistency by assessing the same tasks in both groups without the risk of introducing time-related or carryover effects that can occur in a within-subject design [128]. The mixed-methods design allows data to be analyzed from two different viewpoints: self-reported effects from questionnaires can be compared with behavioral effects found during the task assessment. Altogether, the between-subject mixed-methods design helps to ensure methodological consistency, reduces the occurrence of potential biases, and makes it possible to analyze the congruence between qualitative and quantitative behavioral results. To determine the appropriate sample size for the current study, we conducted an a priori power analysis using G*Power 3.1.9.6 software. Given the novel and highly granular nature of our dependent variable (i.e., an angular error instead of forced choice tasks) compared to previous work, we assumed a medium effect size (r = 0.5) for a Wilcoxon rank sum test. A type I error rate of 0.05 and a power of 0.8 were chosen. Based on these assumptions, the analysis indicated that 56 test subjects would be required for the study.

### 4.1. Participants

Two separate recruitment processes were made for the control and experimental group. Participants were recruited via mailing lists, flyers, and suitable social media platforms. Informed consent was obtained from all participants after they had received a general introduction to the study. The selection of participants was handled based on a general screening questionnaire. Exclusion criteria involved age above 39, current or past substance abuse or addiction, medical abnormalities that could impact cognitive functions, use of psychotropic drugs, and history of neurological conditions or head injury. Moreover, as the experiment required participants to spend multiple hours (in separate sessions) in the VR environment, a history of motion or cybersickness and visual impairments that cannot be corrected by wearing contact lenses led to exclusion from the study. We recruited a total of 62 participants. We initially screened them in a VR test session before the recruitment. Six participants were excluded due to acute motion sickness during this session. Moreover, three participants did not complete the study due to motion sickness and unknown reasons (see Section 2). The final sample contained 53 healthy adults (38 females and 15 males) between 18 and 35 years of age, with a mean age of 23.01 years (*s* = 4.14). The sample can be divided into 27 belt and 26 control participants. These individuals served as participants for a monetary reward or university-internal participation credits. The sample was drawn from the city of Osnabrück and the surrounding area. All participants were fluent in English or German. All study materials were provided in English and German (see Section 4.6). The ethical committee of the University of Osnabrück approved the protocol of the study prior to the recruitment.

### 4.2. Experimental Setup

The same experimental setup was used for all exploration and task assessment sessions to guarantee methodological consistency. The HTC Vive was connected to an experimental computer in order to display the VR environment (see Section 4.2.3). To avoid the connection cables interfering with the participants’ movements, we installed a cable routing system on the laboratory’s ceiling (see Figure 7). Participants were seated on a swivel chair without a backrest. Real-world movement inputs such as walking were not feasible to integrate as the VR environment was too large. Nevertheless, the swivel chair setup required participants to sit upright and allowed smooth body rotations, for instance, when turning. The rotations, in turn, would result in a change in the belt’s signal that adjusts dynamically toward the cardinal north in the VR environment. Moreover, previous research has shown physical rotations to affect navigation similarly to walking [6]. Overall, the laboratory setup allows for exploration that resembles real-world navigation, rotational movements, and viewing behavior while not requiring the possibility of moving freely.

#### 4.2.1. The VR Environment

The VR environment Westbrook with its size of 1 km ${}^{2}$ and its 236 unique buildings, was designed to suit the needs of close-to-real-world spatial navigation research. Westbrook has various streets, ranging from small pedestrian paths to main streets [74]. The movement was limited to paths and streets using a custom-created navigation mesh. We added visual boundaries such as high steps or fences to ensure participants understood intuitively where they could and could not move. As fast movement speed can increase motion sickness [129], movement speed was limited to the virtual equivalent of 5.1 km/h. This resembles the average preferred walking speed of humans aged between 18 and 35 [130]. Up to this limit, participants could dynamically adjust their movement speed by pressing the joystick further or less far in the desired direction. To suit the needs of spatial navigation research in urban environments, the terrain is even, and the city has various features such as parks with trees, bushes and fences, bus stops, cars, bikes, and many more (see Section 4.6). In contrast to a previous study [10], Westbrook was designed to contain no cues regarding cardinal directions. For instance, Westbrook contains a cloudy skybox that does not include an image of the sun. The lighting is handled so that no shadows are displayed. Figure 8 illustrates the layout of “Westbrook” with all 236 buildings enumerated. To avoid confusion, buildings considered part of another building, such as garages, a barn, or a silo, were not counted separately but received the same number as the building they belonged to. Buildings 1–56 are task-relevant buildings, meaning they were used in at least one of the tasks (see Section 4.5). The four orange buildings (see Figure 8) are tall buildings that were placed on the edges of the city to be used as potential landmarks. The 18 buildings marked in blue are residential, while the ones marked in red are stores of various kinds that one would find in a typical small town. All 236 buildings are equipped with custom-size colliders.

#### 4.2.2. The feelSpace Belt

Sensory augmentation was achieved using the feelSpace belt (also see Section 1). The feelSpace belt is a vibrotactile sensory augmentation device in the form of a belt worn around the waist (see Figure 9). It is equipped with sixteen vibromotors evenly spaced around the user’s waist. While it can be used for various purposes, we used it as a tactile sensory augmentation device for cardinal directions. To this end, the belt provided constant vibratory feedback about the cardinal north. Only the vibromotor facing the direction of the cardinal north was activated. This can be achieved through a compass integrated into the control unit of the device (see Figure 9). The smartphone app ’belt control’ provided by the company feelSpace can be used to change the direction of the belt’s signal, for instance, to indicate cardinal south instead. While test subjects were not allowed to change the settings of the belt, we used this feature to change the belt’s signal from indicating real-world cardinal north to the direction we previously defined as virtual north (see Figure 8). To calibrate the belt to match the direction of the virtual north, we asked participants to align themselves with an arrow facing the virtual north that we displayed in VR on the floor of an empty black space. The experimenter then set the belt signal to match this direction.

#### 4.2.3. Experiment Computer and VR System

The exploration and task assessment sessions were conducted using an Alienware desktop computer with an Nvidia RTX 2080 Ti graphics card (driver version 456.71) and an Intel i9 processor. The system was running Windows 10 (64-bit, build-version 19,042) equipped with 32 GB of RAM. During the VR sessions, frame rates were steady at around 90 frames per second (fps), never dropping below 60 fps. This is sufficient to provide a fluent picture while reducing the risk of motion sickness [131]. The VR environment was presented using the HTC Vive Pro Eye head-mounted display (HMD) for the exploration and task assessment sessions. The VR is presented on two AMOLED displays with a resolution of 1440 × 1600 pixels each. The display refresh rate is 90 Hz, and the horizontal and vertical field of view are specified as 106° and 110°, respectively [132]. Eye tracking was recorded throughout the study using the built-in eye tracker but was not included in the final analysis due to the focus on the results of the behavioral tasks. In order to display the VR environment on the HMD, we used SteamVR version 1.16.10. Interaction with the VR environment was possible through Valve Index controllers and an HTC Vive tracker 2.0. The Valve Index controllers are compatible with the HTC Vive VR system. Unlike the regular HTC Vive controllers, they incorporate a joystick for more intuitive movement controls. Additionally, the Valve Index controllers support hand and finger tracking, which is a feature that we implemented to display target locations in the wayfinding task (also see Section 4.5.4). The Vive tracker 2.0 is compatible with the remaining setup and can be used to accurately track the position and orientation of, for instance, the participant’s body. In our study, we mounted the tracker to the chest of the participants (also see Section 4.2.1). To ensure high data quality, we used four HTC base stations 2.0 to track the HMD, the controllers, and the tracker from four sides. Using outside-in tracking, participants’ movements during the experiment can be measured with sub-millimeter accuracy. Overall, the system specifications were sufficient to display the city fluently (>60 fps) at a maximum level of detail and using the maximum render distance. Moreover, the controls aim to make the movement as intuitive and realistic as possible (see Section 4.5), and the precise tracking allows for exact measurements of body movements and rotation.

### 4.3. Questionnaires

We administered three questionnaires during the study to measure training progress and perceived changes to spatial cognition. During the training phase, participants filled in two self-developed questionnaires: one daily and the other one weekly (see Appendix A). These questionnaires aimed to monitor participants’ training progress, potentially occurring technical issues, and observed behavioral changes. For instance, participants were asked to track their daily activities with the belt, their active training time, and their daily belt-wearing duration. We used the responses to these self-report questionnaires to calculate participants’ daily belt-wearing duration as well as active training time (i.e., running, or cycling) with the augmented sense. Control participants did not participate in the training phase. Therefore, they were also not required to answer the daily and weekly questionnaires. Before the first VR session, we assessed the FRS questionnaire [68] in both groups. The FRS is a self-report measure for the use of spatial strategies during everyday life. It consists of 21 questions, each assigned to one of the three scales: egocentric/global, survey, and cardinal directions [68]. The FRS measures participants’ likelihood to apply spatial strategies related to egocentric/global knowledge, survey knowledge, or cardinal directions, respectively. The aim was to compare the FRS results of both groups to analyze whether the groups were different regarding their preferred use of spatial strategies at the start of the experiment. Finally, after completing their last task assessment session, the belt group filled in the FRS questionnaire again. The aim was to compare the preferred spatial strategies of the belt group at baseline with their preferred spatial strategies after several weeks of training with the augmented sense.

### 4.4. VR Exploration Phase

Participants in the VR experiment were required to explore the city of Westbrook, which covered an area of about 1 km^2^ and consisted of 236 unique buildings, including 56 task-relevant buildings with street art on their walls (see buildings 1–56 in Figure 8). The participants were instructed to take photos of any building they encountered with street art, as this would help them become familiar with the buildings that would later be used as target stimuli in the tasks during the assessment phase (for exact task instructions, see Section 4.6). Furthermore, the assignment served as a motivation for participants to explore. The exploration phase consisted of five thirty-minute long sessions in VR, with the belt group wearing the feelSpace belt indicating cardinal north. In contrast, the control group received no information about cardinal directions. Before entering the VR, participants were provided with a movement tutorial to understand the movement controls, which involved using a joystick to move relative to the participant’s facing direction. Participants could take a photo by pressing the trigger button of the controller twice in rapid succession, which was stored as a screenshot. The exploration sessions were repeated five times, with at least 6 h but no more than three working days in between the sessions to ensure methodological consistency.

### 4.5. Task Assessment Phase

Next, participants performed two task assessment sessions. This was to reduce the risk of cybersickness or other known side effects of extended VR exposure. The first session lasted about 60–90 min. In this session, we assessed the pointing north task, the pointing to building locations task, and the building placement task. Player movement was disabled during all tasks in the first task assessment session. This means participants were unable to perform directional movements but could rotate their head and body as usual. We did this to ensure that all participants would have the same visuospatial cues at their disposal. In the second test session, we assessed the wayfinding task with for about 30–50 min. To ensure that participants had fully understood their respective tasks, test trials were provided for all experimental tasks, and participants were only allowed to start the experiment after demonstrating they had fully understood the task and the controls at hand.

#### 4.5.1. Pointing North

We conducted the pointing north task with the feelSpace belt switched off, as belt participants would have otherwise merely had to point into the same direction as the feelSpace belt. Moreover, the control group was left unaware of the virtual direction of the north throughout the experiment to ensure symmetry and completeness with regard to the research design and to test that the VR environment contained no visible direct or indirect identifiers of the direction of cardinal north as intended. In the pointing north task, participants were instructed to point toward the virtual north (see Figure 8). To this end, they were placed in front of the 56 task-relevant buildings (1–56 in Figure 8) in a random order. Pointing was achieved by pointing with a controller (left or right based on personal preference). When pointing the Valve Index controller into a particular direction, a green beam was displayed in the VR to show participants where they were pointing (see Figure 7). Participants moved the controller and thereby controlled the beam to match their intended pointing direction. When participants decided, they could log in their response by pressing the trigger button once. Thereby, the green beam and the corresponding pointing vector could be frozen. Next, participants could confirm the pointing direction by pressing the trigger button again or cancel their selection and point again by pressing the “B” (quick menu) button. Each trial had a maximum duration of 30 s. A timer counting backwards from ten was displayed at the bottom of the screen during the last ten seconds. If no response had been recorded after thirty seconds, the trial timed out and the program advanced to the next trial. After completion or timeout of one trial, a black screen was displayed while participants were placed in front of the next task-relevant building. In order to avoid that participants identified cardinal north correctly once and subsequently only had to point in the same direction, they were rotated away from their previous facing direction in varying degrees between trials. The rotation degrees were randomly selected from a list of incremental angles ranging between −180 and 180 degrees (for data and scripts, see Section 4.6). All 56 unique trials were repeated once, resulting in 112 pointing north trials.

#### 4.5.2. Pointing to Building Locations

Belt participants wore the feelSpace belt during this and all subsequent tasks. In the pointing to building locations task, participants were instructed to point toward a building location displayed in the top middle of the HMD. They were placed at random in front of one of eight task-relevant buildings that were strategically selected for this task based on their categories and locations (see buildings 1–8 in Figure 8). One of the remaining seven building locations was randomly selected as the target location and displayed in the top middle of the screen. After participants pointed toward the target building location, the trial was completed. In the following six trials, participants were required to point toward the remaining six target locations in a randomized order. When participants completed all seven trials from the first starting location, they were teleported to the next randomly selected location out of the remaining seven locations that were not used as starting locations. Participants again pointed toward the seven remaining locations. This procedure was repeated until they had pointed from each of the eight selected locations toward each of the remaining seven, resulting in 56 unique trials. Then, the task was repeated once, resulting in a total of 112 trials. Apart from the points described above, the pointing to building locations task followed the same procedure and incorporated the same control mechanics as the pointing north task (see Section 4.5.1).

#### 4.5.3. Building Placement

In the building placement task, participants were placed at seven different locations in a randomized order. They were required to place four different buildings from each location, also in a randomized order. Overall, participants placed 28 different buildings from 7 different locations. The 28 buildings and the 7 locations were strategically selected based on their location and distance to other task-relevant buildings from the list of 56. More specifically, the seven locations were selected to be as far away from each other as possible but not located on the edges of the city, as this would reduce the placement possibilities. The four buildings that had to be placed from each of these seven locations were selected based on their distance and visibility from the respective location. It was important that the buildings would neither be too far away to place them accurately nor too close to the participant’s location, which could negatively affect the validity of the task. Participants had 1 min and 30 s to place each building. The building that participants had to place was displayed about five meters in front of them. Participants then had to place the building to its correct location using pointing movements and controller inputs. The building placement task required participants to use two Valve Index controllers: one for each hand. While participants could choose their preferred side, we will only explain the controls for a right-handed participant here to provide an example. Similarly to the pointing direction in the pointing tasks, the placement direction in the building placement task could be adjusted by pointing with the right controller. The placement distance could be adjusted by moving the joystick of the left controller forwards to move the building away or backwards to move the building toward the participant’s location. By moving the joystick of the right-hand controller to the right side, the building could be rotated counterclockwise around its axis to adjust the facing direction. To rotate the building clockwise, participants could move the right-hand joystick to the left. All buildings could only be moved around on the floor and could not be lifted into the air. While the view to the correct building locations was occupied by other buildings, buildings and other objects on the way toward the placement location were displayed as see-through. More specifically, by moving the placement building into another object in the city, only the edges of the other object would remain visible, and all other textures within a small ellipse around the placement building would become see-through. Thereby, the participant can see the placement location accurately while still being able to use the object’s skeleton as an orientation point. The task ended when all 28 buildings had been placed.

#### 4.5.4. Wayfinding

For the wayfinding task, participants from both groups were randomly assigned one of two trial orders: regular or reverse. They were instructed to find the most efficient path toward nine different target locations. When they moved the left controller into their field of view, a smartphone was displayed in the respective location on the VR display. Upon starting the experiment, participants were shown a picture of a nearby building that they needed to navigate toward. When they were moving within a 10 m radius of a trigger zone placed in front of the target location, the trigger zone was made visible in the form of a red-colored circle. When participants entered the circle, the trigger zone disappeared, and a new green-colored trigger zone appeared next to it. By entering this trigger zone, a new target building was displayed on the smartphone, and the next trial was initiated. Movement input could be provided by moving the joystick of the right controller in the respective direction. Each trial had a maximum duration that was relative to the fastest time it could take to complete the route. When the maximum duration was reached, a green line was displayed on the floor, guiding the participant to the building displayed on the smartphone. Upon reaching the destination, the subsequent trial was started automatically, and the next target location was displayed. Excluding the test trial, there were a total of 9 trials that all participants underwent.

### 4.6. Statistical Analyses

A significance level of *p* < 0.05 was established for all statistical tests. As a method of analysis, we used a Wilcoxon rank sum test for independent samples to analyze pre-existing differences in the use of spatial strategy (FRS questionnaire) between the two groups. To evaluate changes within the belt group, we used a Wilcoxon sign rank test for dependent samples to compare the preferred spatial strategy used before and after the training. Results from the daily and weekly questionnaires were aggregated to the group level but not analyzed further. As the results from all four experimental tasks were heavily skewed and therefore violated the normality assumption of most parametric tests, we used non-parametric statistical tests. Wilcoxon rank sum tests were performed to compare the two groups’ accuracy in the pointing north task, the pointing to building locations task, the building placement task, and the wayfinding task.

## Figures and Tables

**Figure 1 brainsci-13-00720-f001:**
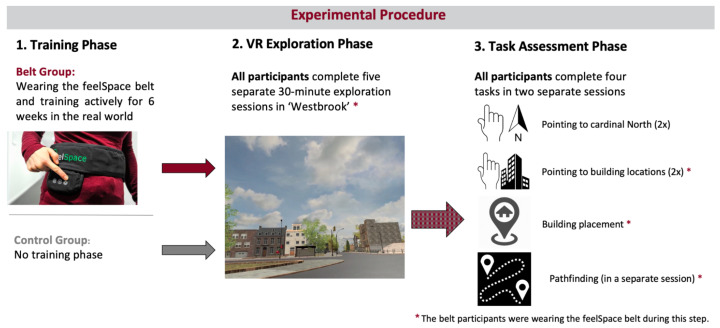
Overview of the experimental procedure. The belt group wore the augmented sense of cardinal north for six weeks during their everyday life before the VR exploration phase started. All participants engaged in five thirty-minute exploration sessions and finally two test sessions in the urban VR environment ‘Westbrook’.

**Figure 2 brainsci-13-00720-f002:**
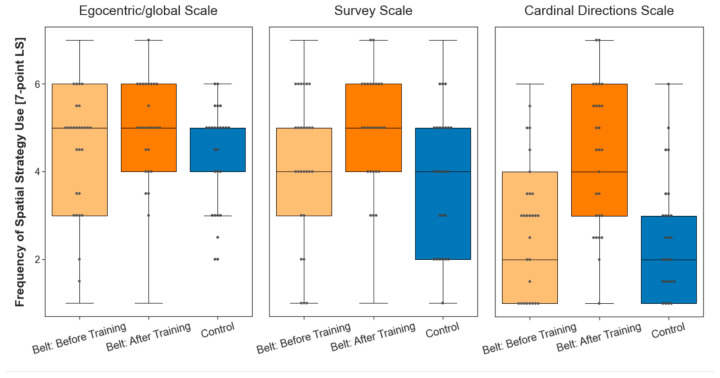
Boxplot of the reported use of egocentric/global orientation, survey, and cardinal spatial navigation strategies of belt participants before and after training as well as in the control group. The gray dots inside the boxplot represent the median pointing error of each participant in the respective group. The whiskers mark the upper and lower extremes, and the vertical lines outside the box mark the upper and lower quartile. While the reported use of spatial strategies is comparable between the control group and the belt group before they engaged in the training, the reported use of spatial strategies after training with the augmented sense for six weeks in real life increased across all three scales.

**Figure 3 brainsci-13-00720-f003:**
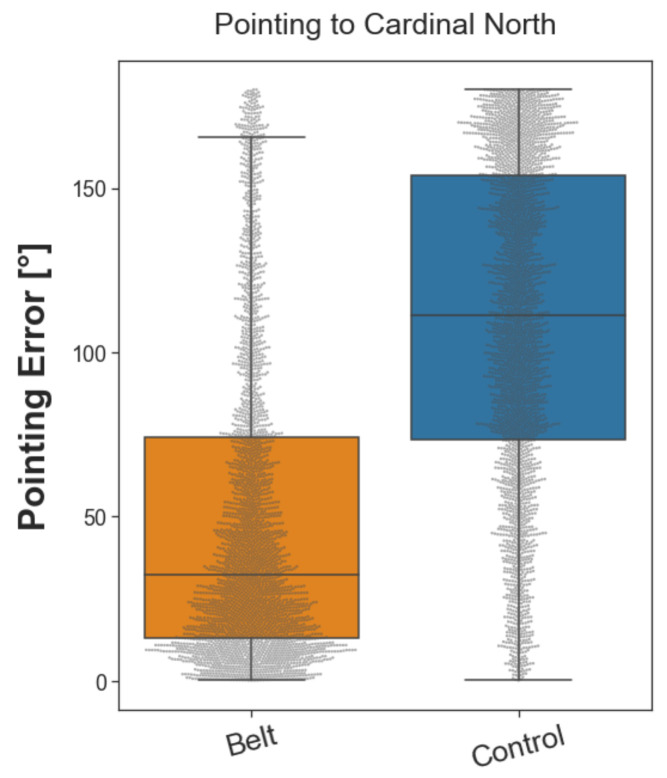
Boxplot of pointing error in the ‘pointing north’ task. The gray dots represent the pointing errors of individual trials. The dots above the upper bound of the boxplot for the belt group represent outliers. The vertical lines outside the box mark the upper and lower quartile, and the whiskers mark the upper and lower extremes. Overall, participants from the belt group had a lower median pointing error (32.37°) than participants from the control group (111.31°).

**Figure 4 brainsci-13-00720-f004:**
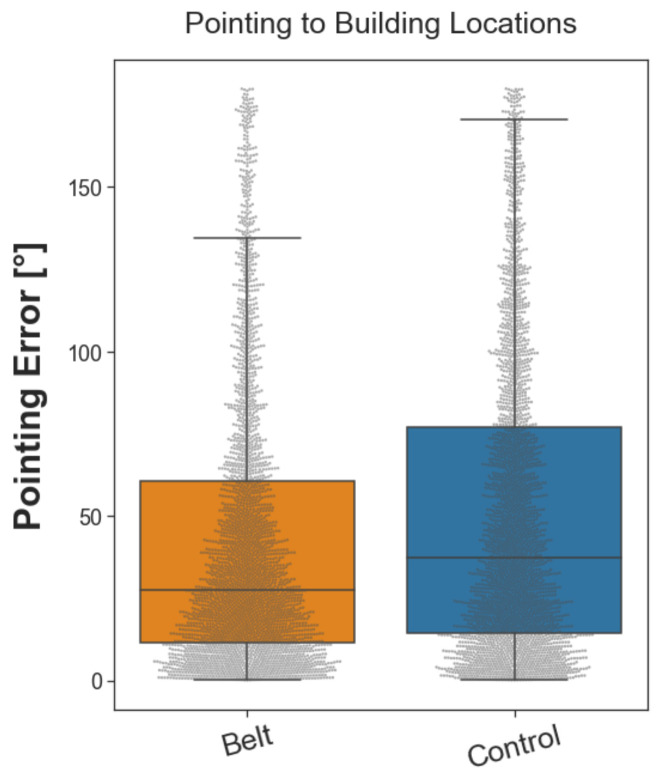
Boxplot of pointing error in the ‘pointing to building locations’ task. The gray dots represent the pointing errors of individual trials. The dots above the upper bound of the boxplot represent outliers. The vertical lines outside the box mark the upper and lower quartile, and the whiskers mark the upper and lower extremes. Although two participants in the belt group had the highest median pointing errors in the sample, participants from the belt group demonstrated a lower median pointing error (27.58°) than participants from the control group (37.41°).

**Figure 5 brainsci-13-00720-f005:**
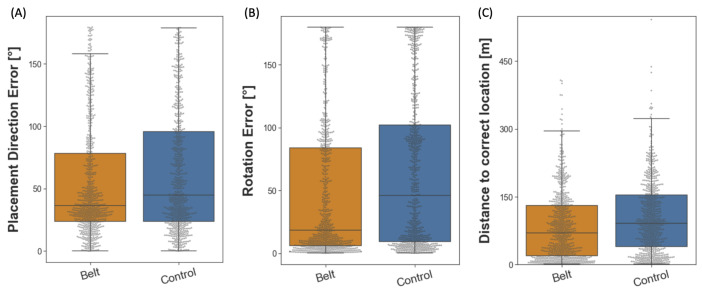
Performance in the building placement task divided by group. The gray dots represent the pointing errors of individual trials. The vertical lines outside the box mark the upper and lower quartile, and the whiskers mark the upper and lower extremes. The dots above the upper whiskers represent outliers. (**A**) Illustrates that the belt group (*Mdn* = 36.60°) placed the buildings more accurately in the correct direction than the control group (*Mdn* = 44.69°). (**B**) Displays that participants from the belt group performed more accurately (*Mdn* = 18.35) in rotating the house to its correct facing direction than control participants (*Mdn* = 45.98°). (**C**) Shows that the belt participants (*Mdn* = 69.97 m) placed the houses closer to their original location than controls (*Mdn* = 91.67 m).

**Figure 6 brainsci-13-00720-f006:**
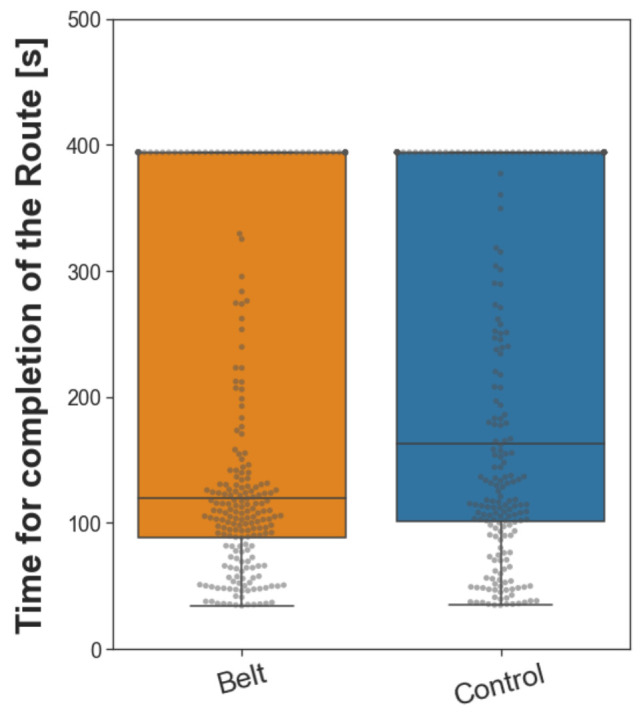
Performance in the wayfinding task separated by group. The gray dots represent the pointing errors of individual trials. The vertical lines outside the box mark the lower quartiles and the whiskers mark the lower extremes. As timed-out trials were assigned the duration of the longest route, there is a ceiling at 394 s. The belt group (*Mdn* = 119.50 s), overall, performed better and reached their target destination quicker than the control group (*Mdn* = 163.39 s).

**Figure 7 brainsci-13-00720-f007:**
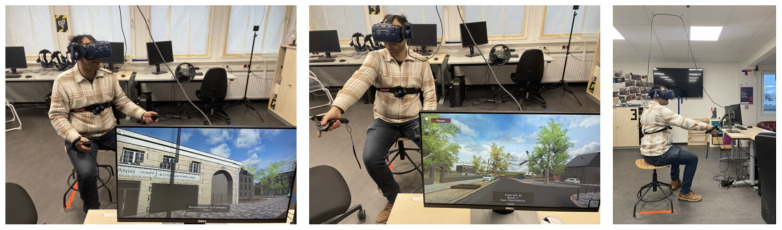
Picture of the laboratory setup. Staged pictures (left to right) of a participant performing a VR exploration session and the pointing north task. The picture on the right shows the experimental setup from the side, including the swivel chair and cable management system.

**Figure 8 brainsci-13-00720-f008:**
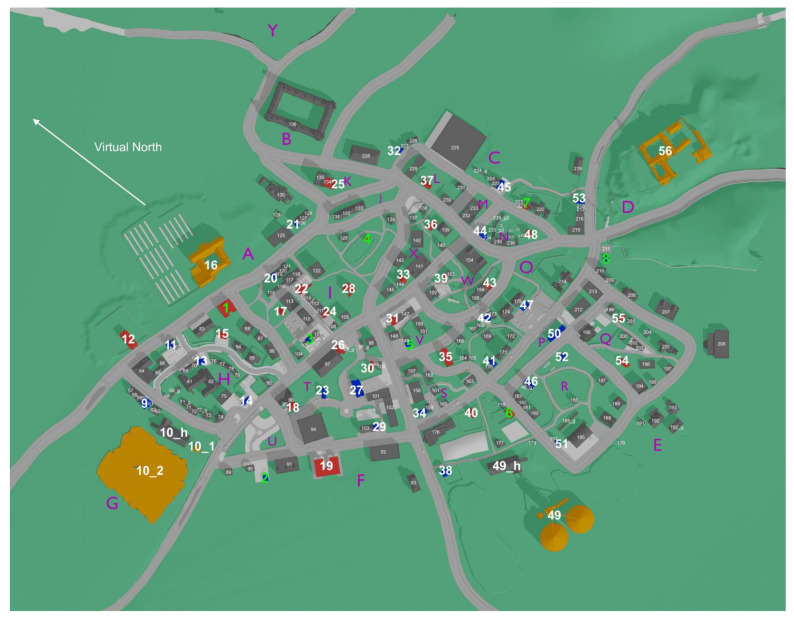
Map of the VR environment “Westbrook” displaying the city’s layout with its 236 unique buildings enumerated. The direction of the virtual north is displayed as a vector in the top left corner.

**Figure 9 brainsci-13-00720-f009:**
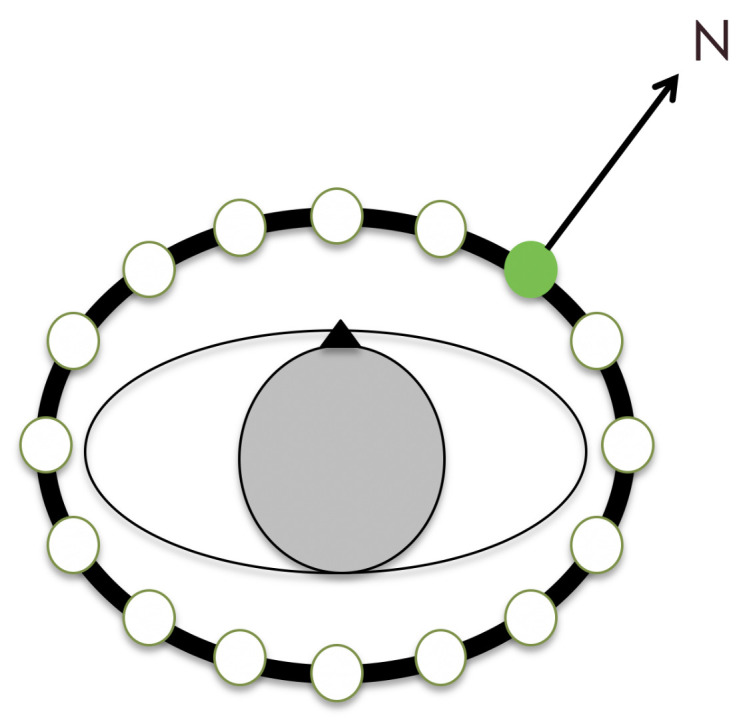
Working mechanism of the feelSpace belt indicating cardinal north or virtual north. By default, the belt indicates cardinal north; the direction of the signal can be adjusted to represent the direction of the virtual north using the ‘belt control’ app.

## Data Availability

All data and analysis scripts are available at https://osf.io/32sqe/ [133]. Additionally, the repository contains the project documentation and the VR exploration and VR task assessment builds for replication or incorporation in future experiments.

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
