# Peer review of "Improved Spatial Knowledge Acquisition through Sensory Augmentation"

_brainsci, 2023, doi:10.3390/brainsci13050720_

Round 1
Reviewer 1 Report
The manuscript titled “Improved Spatial Knowledge Acquisition through Sensory Augmentation” proposes an investigation on the effects of a sensory augmentation device for cardinal directions on different facies of spatial knowledge during navigation. Authors recruited and trained a group of 27 young adults for six weeks with the feelSpace belt, an augmented sense for cardinal directions. A control group of 26 participants did not receive the device nor the associated training. Then all participants explored the Westbrook VR environment for two and a half hours over five sessions, and subsequently were assessed in four immersive virtual reality tasks measuring cardinal, route, and survey knowledge. Results showed that the experimental group (“the belt group”) obtained better scores in cardinal and survey knowledge, measured in pointing accuracy, distance, and rotation estimates. Also route knowledge was positively affected, although with smaller effect sizes, as well as the use of spatial strategies, which had a comparable degree at baseline. Authors discussed their results in light of previous literature, highlighting strengths and weaknesses of their study, and giving hints for further research.
I carefully read the manuscript, and I think it may be of interest for the readers of Brain Sciences. The manuscript is very well-written and properly addresses the interesting issue of augmented senses and spatial learning. Although the present study has been conducted on a sample of young people, it led to interesting results which needs to be replicated in other age cohorts (e.g., healthy elderly people) as well as in special populations (e.g., poor orienteers, people with Mild Cognitive Impairment, et cetera). The paradigm as well as the tool employed seem to be very promising and could also led to promote aided-navigation and orientation intervention protocol in real-world settings. It was a pleasure for me to read the manuscript. The introduction section and the aims of the study are clear and detailed. The methodology is very rigorous and accurate, as well as the explanations provided in the discussion section. I also appreciated the alternative explanations as well as the discussion of limitations, which are often neglected especially when VR environments are used for research purposes. I have no further considerations or remarks.
Author Response
Dear Editorial Board Members and Reviewers,
We are grateful for your time and expertise in reviewing our manuscript, "Improved Spatial Knowledge Acquisition through Sensory Augmentation," submitted to Brain Sciences. Your constructive critiques were insightful and helped us to improve the manuscript. Below, we itemize how we incorporated them into the manuscript. We will present all revisions marking them with a different color on the paper itself and adding the page and line numbers as a reference for the implemented changes on the revision letters.
Response to Reviewer 1
Comment 1: I carefully read the manuscript, and I think it may be of interest to the readers of Brain Sciences. The manuscript is very well-written and properly addresses the interesting issue of augmented senses and spatial learning. Although the present study has been conducted on a sample of young people, it led to interesting results which needs to be replicated in other age cohorts (e.g., healthy elderly people) as well as in special populations (e.g., poor orienteers, people with Mild Cognitive Impairment, et cetera). The paradigm as well as the tool employed, seem to be very promising and could also lead to promoting aided navigation and orientation intervention protocol in real-world settings. It was a pleasure for me to read the manuscript. The introduction section and the aims of the study are clear and detailed. The methodology is very rigorous and accurate, as well as the explanations provided in the discussion section. I also appreciated the alternative explanations as well as the discussion of limitations, which are often neglected especially when VR environments are used for research purposes. I have no further considerations or remarks.
Addressed (added / replaced paragraph): We appreciate your generous recount of the paper. We agree that further studies in other age cohorts as well as special populations are needed. A recently published study [49] used simple outdoor tests on a sample of blind people and found that using the belt reduced discomfort and increased confidence while navigating. Moreover, participants reported feeling safer wearing the belt in different outdoor scenarios. We added a short discussion including references to this study in addition to the work of other labs in the revised manuscript (added paragraph on page 14, lines 509-526).
Reviewer 2 Report
The manuscript describes a study testing whether augmenting the spatial sense of knowing where north is improves spatial ability when navigating in a VR environment. The study found support, of various strength, that this augmentation increases spatial ability.
The strengths of the study are that it is a fairly well-controlled experiment, with extensive preparation (six weeks of pre-training), adequate sample sizes, and a good coverage of testing spatial abilities. Weaknesses are some methodological issues, most notably description of group assignment, missing sample size justification, issues with the pointing north task, and possible a confounding factor of wearing a belt vs. not wearing a belt. An important weakness is that a theoretical explanation of why the belt would improve spatial ability is missing.
Detailed comments:
Introduction
Overall, the introduction introduces and motivates the current study well, and is well-referenced (there are quite a number of self-references, but that is to be expected given the topic and the use of the particular technology).
A theoretical reflection: Isn’t the embodied perspective of perception and action (expressed in the first paragraph) incompatible with the idea of internal mental maps and other mental representations (introduced in the second paragraph)? This may reflect a general lack of focus on theory throughout the manuscript – explanations for effects are not discussed, and therefore there is no real conflict between embodiment and representationalist views in the manuscript.
A minor methodological reflection: “Moreover, the exact layout of the city but also the weather and light conditions, as well as the number of people in the streets, cannot be controlled to be comparable across participants.” (lines 102-103): I would say factors such as these would be handled with balancing using block randomization (although it increases unwanted variability in the measurements).
The experimental procedure is summarized in an informative figure (Figure 1) that is easy to understand.
Materials and Methods
The sample size is not justified (e.g., Lakens, 2022). An a priori power calculation to estimate sample size is missing.
Lakens, D. (2022). Sample size justification. Collabra: Psychology, 8(1), 33267. https://doi.org/10.1525/collabra.33267
It is not stated that random assignment was used when forming the two groups. I suspect it was, but if groups were formed, for instance, by participant’s choice, that would be a critical confounding factor.
A note on clarity: I could not understand the sentence: “This ensures that no objects that should be visible are not displayed or with less detail.” (lines 631-632)
4.5.1 Pointing North:
For the pointing north task, as far as I understand, the belt group had previously received information where the virtual north was, so they could just use landmarks to identify the virtual north in the absence of the augmentation of the belt.
I’m having a hard time understanding the pointing north task. Isn’t it a meaningless task for the control group to point to virtual north when they have never received information about it?
How was the lighting done in VR? Was the sun accurately positioned in relation to the time of day? The position of the sun may be a clue when performing all the tasks.
Besides the pointing north task, I believe the other tasks are appropriate measures of spatial knowledge.
Results
Writing suggestion: The results sections contain a fair amount of methodological details. This is in a way informative, but it also makes the results section more difficult to overview. Some information is also duplicated from the methods section. I believe one reason for why a need is perceived for methodological details is that the Materials and Methods section is last in the paper. If it were moved to before the Results (as the standard IMRAD structure), the need for methodological details in the Results sections would be reduced, making it more succinct.
Figure 3: It is remarkable that the errors of the control group as so tightly centered around 100 (+/- 20) – wouldn’t it be expected that these errors would spread fairly evenly across 0-180 degrees? It seems something made the control group think that the virtual north was around 100 degrees away from where it actually was (could it be clues from the lighting?).
Discussion and conclusion
A power analysis is mentioned on lines 229-330, but no details of this are presented.
How do we know that it was not the exploration phase in VR (with indication of virtual north for the belt group), and not the six weeks training, that had the effect on the tasks? Perhaps the case of not using a belt at all, just being informed where north was (e.g., symbolically) may have had the same result?
The belt group did so many other things than just augmenting a sense: the physical wearing of the belt, exercise, etc. A more critical comparison of only the augmented sense would compare two groups, both wearing belts and doing identical training, but where the sense was augmented in one group only.
I wonder what the six-week training phase achieved: in other words, what is the explanation for the improved spatial ability?
Author Response
Dear Editorial Board Members and Reviewers,
We are grateful for your time and expertise in reviewing our manuscript, "Improved Spatial Knowledge Acquisition through Sensory Augmentation," submitted to Brain Sciences. Your constructive critiques were insightful and helped us to improve the manuscript. Below, we itemize how we incorporated them into the manuscript. We will present all revisions marking them with a different color on the paper itself and adding the page and line numbers as a reference for the implemented changes on the revision letters.
Response to Reviewer 2
Comment 1: A theoretical reflection: Isn’t the embodied perspective of perception and action (expressed in the first paragraph) incompatible with the idea of internal mental maps and other mental representations (introduced in the second paragraph)?
Addressed (deleted line and added theoretical clarification): This is a really good point that triggered discussion amongst the authors. In order to avoid inconsistency, we have removed all references to mental maps (rephrased text on page 2, line 37) and to clarify our stands on how spatial learning is acquired in the context of enaction theory, we have included the references [23, 24] for readers who want to further review this theoretical position (added text page 2, in lines 38-39). The present manuscript, however, focuses on the behavioral benefits of sensory augmentation and is not the right place for a full-blown discussion of this issue.
Comment 2: Methodological reflection: “Moreover, the exact layout of the city but also the weather and light conditions, as well as the number of people in the streets, cannot be controlled to be comparable across participants.” (lines 102-103): I would say factors such as these would be handled with balancing using block randomization (although it increases unwanted variability in the measurements).
Addressed (deleted text): We agree, in theory, it could be really difficult but possible to create a balanced experimental design in a real-world scenario by using random block assignment if one is willing to sacrifice homogeneity of exposure amongst participants. We removed that statement (deletion on page 3, lines 105-107) since its purpose was to convey the argument for real-world scenarios being hard to balance, and this point has already been raised in lines 102-103.
Comment 3: The sample size is not justified (e.g., Lakens, 2022). An a priori power calculation to estimate sample size is missing. Lakens, D. (2022). Sample size justification. Collabra: Psychology, 8(1), 33267. https://doi.org/10.1525/collabra.33267
Addressed (added section): Thank you for pointing it out. We have now added this information in the materials and methods detailing how we calculated the a priori power analysis (added text on page 15, lines 575-581).
Comment 4: It is not stated that random assignment was used when forming the two groups. I suspect it was, but if groups were formed, for instance, by participant’s choice, that would be a critical confounding factor.
Addressed (added text for methodological clarification and added discussion): Due to significant differences in subject involvement and varying timelines, a random subject assignment was not feasible. Instead, two separate calls were made, potentially introducing biased selection/assignment to groups. To address this concern, we administered the Questionnaire of spatial strategies (FRS) to belt subjects both before and after training, and to control subjects once. Our analysis revealed no systematic differences between belt and control subjects prior to training (page 4, lines 157-158), but significant differences were observed after training (page 5, lines 185-187). While a confounding bias cannot be entirely ruled out, our testing does not give an indication of such a bias.
In order to underline this limitation to the readers, we now have addressed it in:
-
- the Methods section (text added on page 15, line 583)
- the Discussion (replacement pages 11-12, lines 383-391):
Comment 5: A note on clarity: I could not understand the sentence: “This ensures that no objects that should be visible are not displayed with less detail.” (lines 631-632).
Addressed (deleted text): The sentence aims to clarify that the presentation of objects in the city is relative to their proximity to the participant, thereby maintaining an appropriate level of detail. Nevertheless, the sentence prior already states, “Overall, the system specifications were sufficient to display the city fluently (>60fps) at a maximum level of detail.” which conveys the same idea. Therefore, we have deleted that sentence (on page 18, lines 686-687).
Comment 6: 4.5.1 Pointing North: For the pointing north task, as far as I understand, the belt group had previously received information where the virtual north was, so they could just use landmarks to identify the virtual north in the absence of the augmentation of the belt. I’m having a hard time understanding the pointing north task. Isn’t it a meaningless task for the control group to point to virtual north when they have never received information about it? How was the lighting done in VR? Was the sun accurately positioned in relation to the time of day? The position of the sun may be a clue when performing all the tasks. Besides the pointing north task, I believe the other tasks are appropriate measures of spatial knowledge.
Addressed (added text for methodological clarification and added discussion): Thank you for helping us to clarify our reasons for performing the pointing to north task in both groups. In the belt group, the Pointing North task goal was to assess whether participants had acquired spatial knowledge of cardinal directions and could explicitly retrieve this information when the belt was inactive. In contrast to a previous study [12], the virtual environment in the current study, therefore, contained no indirect cues regarding cardinal directions i.e. there were no shadows, and we used diffused lighting (added text on page 16, lines 632-635). The control group never received information related to cardinal directions, and the pointing to North task might be regarded as ‘random pointing’ for it. For symmetry and completeness, we kept this task for the control group (added on page 19, lines 747-748). Further, it is a safeguard against any involuntary cues on cardinal directions that might have entered the VR during the design phase. This might be expected but is not given. Finally, we added a comment in the appropriate place in the result section (added on page 6, lines 225-227).
Comment 7: Writing suggestion: The results sections contain a fair amount of methodological details. This is in a way informative, but it also makes the results section more difficult to overview. Some information is also duplicated from the methods section. I believe one reason why a need is perceived for methodological details is that the Materials and Methods section is last in the paper. If it were moved to before the Results (as the standard IMRAD structure), the need for methodological details in the Results sections would be reduced, making it more succinct.
Addressed (revised, moved and deleted text). We appreciate your suggestion. Given that our method section is quite lengthy, as it contains all the necessary details for technical replication, we opted to use the IRDAM format to enhance readability and provide a succinct summary of the important methodological information in the results section. We believe that this approach will make it easier for readers to comprehend the core methods used in our study while still maintaining the necessary rigor and detail. However, we acknowledge that duplicating information is unnecessary, and as a result, we have removed any redundant information from the results section as follows:
-
- Section 2.1 Baseline comparison: deleted text on page 4, lines 149-156
- Section 2.2. Training Phase of the Belt Group: deleted text on page 4, lines 162-165
- Section 2.3. ​​VR Exploration Phase: the information that was unique to the VR exploration phase in Section 2.3 was transferred to Section 4.4 (page 19, lines 714-729), which is the Methods section. The remaining content in Section 2.3 was removed from pages 5-6, lines 192-207.
- Section 2.4 Task Assessment Phase delete text on page 6, lines 216-218.
Additionally, we have included links in the results section to relevant subsections in the method section for convenient access to detailed methodological information as follows:
-
- Section 2.1 Baseline comparison: added link on page 4, line 149.
- Section 2.4 Task Assessment Phase: added link on page 6, line 209
- Section 2.4.4. Wayfinding added link on page 9, line 288
Comment 8: Figure 3: It is remarkable that the errors of the control group are so tightly centered around 100 (+/- 20) – wouldn’t it be expected that these errors would spread fairly evenly across 0-180 degrees? It seems something made the control group think that the virtual north was around 100 degrees away from where it actually was (could it be clues from the lighting?).
Addressed (changed Figures 3, 4, 5, 6): This feedback proved valuable as it prompted us to reassess our data plotting approach. Initially, we were displaying errors as an aggregate unit by using the median errors of all trials across one participant, inadvertently creating a biased depiction of the actual individual distribution of errors. In response, we have revised Figures 3-6 to depict the distribution of individual errors across the 0-180 degree range (see pages 7, 8, 9, and 10 respectively). Each gray dot now represents the outcome of an individual trial of a participant within the respective group. Moreover, we have adjusted the figure captions accordingly (see Figure captions 3-6). Upon closer examination of the individual trial data, it is clear that errors are more or less equally distributed in the control group, while they are skewed towards zero in the belt group.
Comment 9: How do we know that it was not the exploration phase in VR (with an indication of virtual north for the belt group), and not the six weeks of training, that had an effect on the tasks? Perhaps the case of not using a belt at all, just being informed where north was (e.g., symbolically) may have had the same result? The belt group did so many other things than just augmenting a sense: the physical wearing of the belt, exercise, etc. A more critical comparison of only the augmented sense would compare two groups, both wearing belts and doing identical training, but where the sense was augmented in one group only. I wonder what the six-week training phase achieved: in other words, what is the explanation for the improved spatial ability?
Addressed (rephrasing for precision and added discussion): Good point and well taken. In the present study, we demonstrate that the combination of training with the belt in natural environments, and exploring the VR with the belt is sufficient for the behavioral benefits. In the revised manuscript we carefully rephrase the respective statements. Further, previous research has shown that the training phase by itself and using the augmented sense during VR exploration (without previous training) is not sufficient for cognitive and perceptual changes to occur. However, whether the tested combination of training with the belt and using the belt in VR is the minimal sufficient, i.e. necessary set remains to be shown. In the revised manuscript we emphasize these points (rephrased in lines on page 7 line 252, page 10 lines 324-326, page 12 lines 418-419, page 13 lines 440-442, page 13 lines 468-469, page 13 line 473, page 13 line 479). Finally, we added a more elaborate discussion of the points mentioned above (on page 13, lines 482-502).
Reviewer 3 Report
In the paper “Improved Spatial Knowledge Acquisition through Sensory Augmentation" the authors investigated the relationship between sensory augmentation and visuospatial navigation skills. To this aim the authors asked participants to wear a belt able to increase their sense of cardinal directions and explore a VR environment. The study is timely and sound. There may be room for improving the discussion in order to link the present results to a broader framework.
1) Abstract – Although being very clear, that abstract does not seem to mention the possible implications of the study, beyond observational/descriptive results. It is suggested to add a sentence about possible applications.
2) Discussion - What should be learnt by a reader from another domain? For example, the possible exploitation of the present results beyond basic science might benefit from the consideration that the methods developed in the present study could be used to ameliorate (i) clinical assessments (e.g. considering the variations in the subdivisions of responsice and non-responsive people to the effects of the belt) and (ii) intervention protocols (e.g. fostering a better focus on possibly customized therapeutic approaches as a function of the individual responsiveness to the belt). These perspectives could specify the possible impact of the present study’s methods and findings, with possible direct links to the development of innovative strategies to face the documented need of boosting sensory restoration techniques in the context of neurorehabilitation (Perruchoud et al 2016, Journal of Neural Engineering). This way it could be clearer how sensory augmentation could improve sensory restoration and, therefore, sensibly advance the techniques to restore proper loops between perception and behavior, extending the impact of the present study to a wide range of conditions characterized by impaired sensorimotor integration.
Author Response
Dear Editorial Board Members and Reviewers,
We are grateful for your time and expertise in reviewing our manuscript, "Improved Spatial Knowledge Acquisition through Sensory Augmentation," submitted to Brain Sciences. Your constructive critiques were insightful and helped us to improve the manuscript. Below, we itemize how we incorporated them into the manuscript. We will present all revisions marking them with a different color on the paper itself and adding the page and line numbers as a reference for the implemented changes on the revision letters.
Response to Reviewer Three
Comment 1: Abstract – Although being very clear, the abstract does not seem to mention the possible implications of the study, beyond observational/descriptive results. It is suggested to add a sentence about possible applications.
Addressed (added text): Thank you very much for your insightful review of our article. We agree that possible applications are an important topic. Therefore, in the abstract, we included an additional sentence addressing the issue (page 1, lines 14-16). Furthermore, we revised the discussion along these lines following your second comment (see below).
Comment 2: Discussion - What should be learned by a reader from another domain? For example, the possible exploitation of the present results beyond basic science might benefit from the consideration that the methods developed in the present study could be used to ameliorate (i) clinical assessments (e.g.considering the variations in the subdivisions of responsible and non-responsive people to the effects of the belt) and (ii) intervention protocols (e.g. fostering a better focus on possibly customized therapeutic approaches as a function of the individual responsiveness to the belt). These perspectives could specify the possible impact of the present study’s methods and findings, with possible direct links to the development of innovative strategies to face the documented need of boosting sensory restoration techniques in the context of neurorehabilitation (Perruchoud et al 2016, Journal of NeuralEngineering). This way it could be clearer how sensory augmentation could improve sensory restoration and, therefore, sensibly advance the techniques to restore proper loop between perception and behavior, extending the impact of the present study to a wide range of conditions characterized by impaired sensorimotor integration.
Addressed (added text in discussion): Thank you for this suggestion. Previous research with the feelSpace belt investigated it as an assistive device for blind people [41, 128] and out of the scientific background, a start-up firm named "FeelSpace" was founded. We gladly address this important aspect in the revised manuscript by restructuring section 3.2 Sensory Augmentation can Enhance Human Performance after training (pages 13-14, lines 508-526). Here, we also added the recommended reference which provided us with great new insights about further potential applications in BMIs.
Round 2
Reviewer 3 Report
accept